# The Impacts of Electric Vehicle Growth on Wholesale Electricity Prices in Wisconsin

**Megan Zielke [1,2,\*], Adria Brooks [1,3] and Gregory Nemet [2]** 

[1]   Office of Regional Energy Markets, Division of Energy Regulation and Analysis, Public Service Commission of Wisconsin, Madison, WI 53705, USA; brooks7@wisc.edu

[2]   La Follette School of Public Affairs, University of Wisconsin—Madison, Madison, WI 53706, USA; nemet@wisc.edu

[3]   Department of Electrical and Computer Engineering, College of Engineering, University of Wisconsin—Madison, Madison, WI 53706, USA

[\*]   Correspondence: mmzielke@wisc.edu

**Abstract:** This work explores the impact of the rapid growth of plug-in electric vehicles on wholesale electricity pricing. Understanding electric vehicle impacts on the grid is important for the mid- and long-range planning of transmission owners, distribution utilities, and regional system operators. Current research in electric vehicles considers technology adoption projections and the infrastructure needed to support electric vehicle growth. This work considers how projected electric vehicle growth in the State of Wisconsin would impact the transmission congestion and wholesale electricity pricing in the year 2030. We find minimal impacts on electricity prices (<2%) even under rapid growth assumptions, in which EVs comprise 5% of all vehicles in 2030. The increases seen in hourly locational marginal prices (LMPs) due to projected electric vehicle growth are, on average, less than those seen in annual changes of historic electricity prices in Wisconsin. We do find moderate, relative increases in congestion prices (+16–32%), which could provide an opportunity to align electric vehicle charging schedules with times of low transmission congestion.

**Keywords:** wholesale electricity pricing; location marginal prices; load growth; transmission planning; transmission congestion

## 1. Introduction

The transmission system was built to accommodate delivery of power from generators to loads given a snapshot in time. As both the world's generation mix and load profiles evolve, electric transmission systems must similarly adapt. One indication that new transmission lines are needed is an increase in power flow congestion on existing lines. Concern over how the transmission grid must evolve to accommodate increased electric vehicle charging load has arisen among practitioners.

Transmission owners are tasked with maintaining a reliable electric transmission system and installing transmission infrastructure when necessary. State regulators are tasked with protecting ratepayer interests and ensuring any increase in electricity operational costs are justified by system need. The delicate balance of transmission system planning given changing generation and load is overseen by the Regional Transmission Operators, of which both transmission owners and regulators are participatory stakeholders. In order to plan necessary upgrades to the electric grid, practitioners first test how anticipated changes would strain the existing electric grid. Only then will operators consider what novel control schemes, metering tools, and grid modernization efforts common in the research literature could be applied to address these concerns.

This research was motivated to answer the question: Will the projected growth in Wisconsin plug-in electric vehicles (PEV) increase electric system congestion and require the construction of new transmission lines? To answer this question, we use the grid modeling tools available to operators, assuming no changes to the existing Wisconsin electric grid, and use realistic electric vehicle growth assumptions. This research required access to proprietary electric grid data not commonly available to researchers for security purposes. This work is written to address the concerns of transmission system practitioners, upon which the research community can build.

Current electric vehicle research considers technology adoption projections and the infrastructure needed to support PEV growth [1–3], and a smaller subset of studies have considered the specific impacts on distribution and transmission electric grids [4–6]. There are few studies specific to the Midwestern United States [3], but no existing work for Wisconsin regarding PEV load growth and transmission congestion. This study analyzes how electrification of light-duty vehicles will transform the use of the electric power grid along bulk transmission lines. This work considers how projected electric vehicle growth in the State of Wisconsin would impact transmission congestion, using wholesale electricity pricing as a proxy for system impacts. The objective of this research is to create a better understanding of how an emerging PEV market can impact transmission system need.

The structure of this article is as follows. A review of the current electric vehicles registered in the State of Wisconsin is given in Section 1.1. An overview of the regional electric grid and the many system operators in the State is given in Section 1.2. A discussion of the differences between wholesale and retail electricity pricing, including the regulatory management of prices, is also included in Section 1.2. Section 2 reviews the methods used in this study, namely, estimates of electric vehicle growth, EV charging profiles, and the electricity market simulation tool used. Section 3 describes the wholesale electricity price results for the three electric vehicle growth scenarios. A discussion of the policy implications of the change in wholesale pricing due to EV growth is given in Section 4. Conclusions are presented in Section 5.

*1.1. Electric Vehicles in Wisconsin*

The United States Energy Information Administration (EIA) estimates there are currently 1.7 million plug-in electric vehicles in the U.S. today [7], and there are several organizations with competing future projection estimates of PEV adoption. The Organization of the Petroleum Exporting Countries and Bloomberg, for example, respectively estimate that worldwide electric vehicles will reach 266 million and 559 million by 2040 [8,9]. Vehicle manufacturer investment and electric utility pilot programs show that key actors anticipate this growth [2]. Manufacturers such as General Motors, Ford, Nissan, Jaguar, Porsche, and Volkswagen have been investing billions into research and development, training, and new employees, evidence that the PEV industry is rapidly evolving [10,11].

The transportation technologies discussed will be mainly passenger vehicles, light-duty vehicles, and fleet vehicles registered with the Wisconsin Department of Transportation (WisDOT). There are three main categories of PEVs: hybrid-electric vehicles (HEVs), plug-in hybrid electric vehicles (PHEVs), and battery electric vehicles (BEVs). HEVs use an internal combustion engine (ICE) system along with an electric propulsion system, resulting in improved fuel economy and performance compared to standard ICE vehicles. These improvements occur through regenerative braking, idling, and electric-only drive. HEVs also use gasoline or diesel as fuel and do not require battery charging. PHEVs use an ICE system along with an electric propulsion system and a rechargeable battery. These batteries are charged through an outlet and allow PHEVs to drive extended distances on an electric charge. These vehicles typically run on only electricity until the battery depletes, at which point the ICE engages and then powers the vehicle. BEVs use a rechargeable battery and run on electricity alone. BEVs are charged by plugging into an outlet and can travel between 100 and 335 miles on a single charge, depending on the battery size, make, and model. In this report, plug-in electric vehicles are those vehicles that utilize battery charging: PHEVs and BEVs. HEVs do not use electricity from the grid, therefore they are not applicable to the transmission impacts studied here.

In September 2018, there were just over 2100 plug-in electric vehicles registered with the Wisconsin Department of Transportation [12]. In February 2019, the Alternative Fuels Data Center identified nearly 600 public charging outlets in the State of Wisconsin of all levels [13]. The geographical dispersion of registered electric vehicles and public charging stations are shown in Figures 1a and 1b, respectively. Figure 1c shows the distribution utility jurisdictions in Wisconsin. Unsurprisingly, the location of public charging stations closely aligns with the locations of registered plug-in electric vehicles. Level 1 chargers are standard 110V outlets. Level 2 chargers are 240V chargers like those used in the United States for clothes washing machines. Level 3 chargers are fast DC chargers used only by the long-range BEVs [1].

The highest concentration of EVs occur in urban areas like Milwaukee, Madison, La Crosse, and Green Bay. Wisconsin Electric Power Company, Madison Gas & Electric, and Wisconsin Power & Light are the major investor owned utilities representing these urban areas, as shown in Figure 1c. According to [14], Wisconsin is 29th in the U.S. with a 0.79% market share of existing vehicles within the state in 2018. Sixty percent of these Wisconsin vehicles are plug-in hybrid vehicles (PHEV).

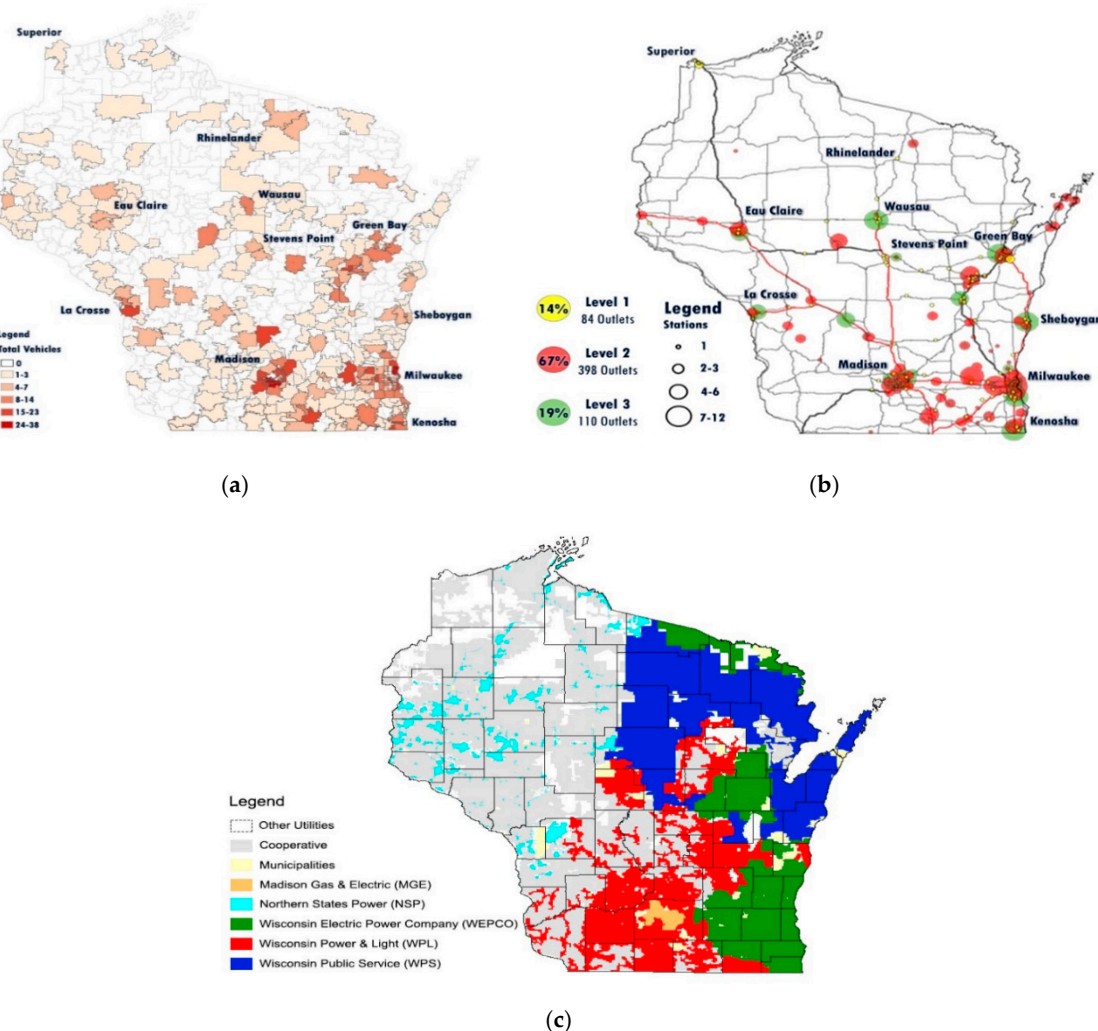

**Figure 1.** (**a**) Electric vehicle (EV) registrations in the State of Wisconsin as of September 2018 by zip code, data from [12]. (**b**) Plug-in Electric Vehicle (PEV) charger locations in Wisconsin by charging level, data from [13]. (**c**) Major Wisconsin utility jurisdictions and corresponding county, data from [15].

*1.2. The Bulk Electric System in Wisconsin*

The Wisconsin transmission grid is overseen by the Midcontinent Independent System Operator (MISO), the Regional Transmission Organization in the middle of the United States. There are five investor-owned distribution utility companies in Wisconsin: Wisconsin Power & Light (WPL), Madison Gas & Electric (MGE), Northern States Power Company (NSP), Wisconsin Energy Power Company (WEPCO), and Wisconsin Public Service (WPS). Areas not served by these distribution utilities are mostly served by small municipalities and cooperatives. NSP operates in both Minnesota and Wisconsin, with the majority of load serving the twin cities of St. Paul and Minneapolis, Minnesota. These entities are shown geographically in Figure 1b.

The price for electricity on the transmission grid is calculated as the cost of servicing the next increment of power demand at a specific electric bus, known as the locational marginal price (LMP) [16,17]. There are numerous electric buses in a utility company's service territory, with the exact number depending on the size of the service territory. The LMP is comprised of three cost components: energy, congestion, and losses. The energy LMP component is uniform across an entire market region, but inefficiency losses and energy congestion on any transmission line can cause nonuniform prices between nodes [17]. Areas with a more expensive congestion LMP component indicate that there is transmission line congestion causing difficulty in delivering electricity to loads in that area, requiring a more expensive generator to be dispatched elsewhere.

If the congestion is significant or present for long periods of the year, practitioners may choose to alleviate the congestion in order to lower annual system operating costs. The Federal Energy Regulatory Commission requires system operators to lower system congestion based on locational marginal pricing [16]. Additionally, system operators make transmission investment decisions—most notably new transmission lines—to manage system congestion and lower wholesale electricity prices. Since 2007, MISO has approved over $328 million in Market Efficiency Projects, new transmission lines meant to alleviate system congestion and bring benefits to customers in the form of lower LMP electricity prices [18].

The locational marginal price is the wholesale price of electricity traded among market participants in the regional real-time energy markets. Retail customers are not typically exposed to the real-time price volatility of wholesale electricity prices, though advances in smart metering technology have renewed a debate that they should be [19]. Distribution utilities set retail electricity rates such that they can recover their costs of trading electricity in the wholesale energy markets over longer periods than the real-time market, as regulated by their state's public utility commission [20]. Load growth is one of many factors in an evolving electric grid that can influence transmission line congestion and changing wholesale electricity prices.

The historic (2012 to 2018) hourly locational marginal prices (LMPs) of all five major service territories are shown in Figure 2, organized both by service territory and by year. Monthly average LMPs for all Wisconsin investor owned utilities are shown in Figure 3. Historic LMP data for each service territory are from MISO's archived markets database, representing ex post hourly LMPs at the utility hub [21]. These historic LMP data will be compared to modeled changes in LMPs due to electric vehicle growth in subsequent sections. Modeled LMPs used later are load-weighted averages of hourly LMPs for every bus in a given service territory.

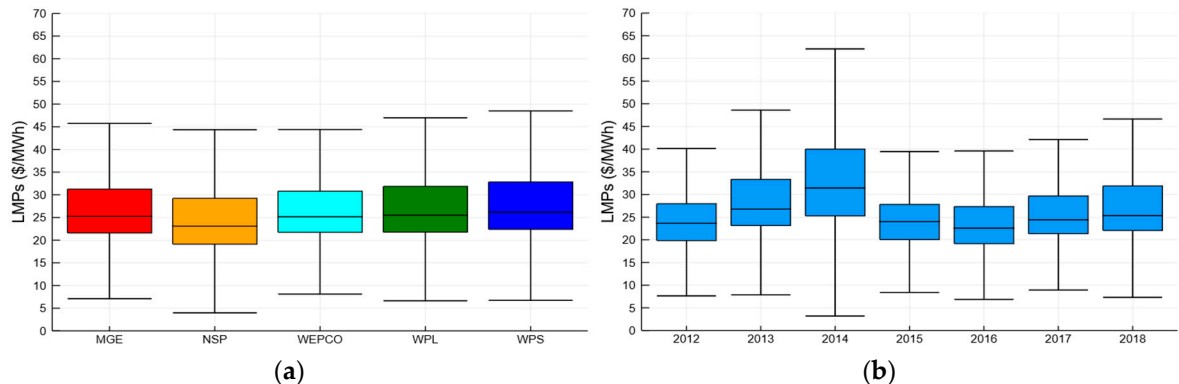

**Figure 2.** Historic, ex post hourly locational marginal prices (LMPs) for five major Wisconsin utilities from 2012–2018, data from [21]. Organized both by utility (**a**) and by year (**b**).

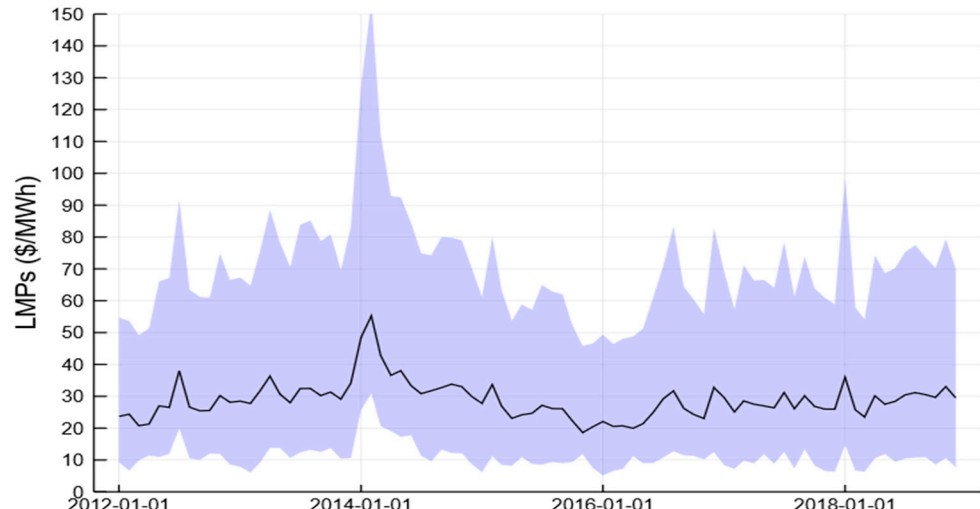

**Figure 3.** Monthly averages of historic, ex post hourly LMPs for all five Wisconsin utilities between 2012 and 2018, data from [21]. Monthly averages are indicated by the black line, with 10th to 90th percentiles shown as the shaded area.

Historic electricity prices on Wisconsin's bulk electric system were very similar among the different utilities. All utilities havd increasing LMPs from 2012 to 2014, a decrease from 2014 to 2016, and then increase again between 2016 and 2018. The percent change in average hourly LMPs from year to year ranged from 1% (2015 to 2016) to 33% (2014 to 2015). The upper Midwest experienced a polar vortex in winter 2013–2014. The extreme cold resulted in larger than average LMP prices for several months. These data show the extent of volatility in LMPs in Wisconsin over the past several years.

## 2. Materials and Methods

We consider how electric vehicle growth would impact the current Wisconsin transmission system in the year 2030. Given that future PEV adoption in Wisconsin is unknown, we consider several different growth scenarios, which serve as reasonable possibly pathways rather than forecasts. We are most interested in changes in transmission congestion and how that would impact wholesale prices for each of Wisconsin's five main utility service territories.

### 2.1. Wisconsin PEV Charging Profiles

We used the WisDOT registration information from 2018 to quantify the number of plug-in electric vehicles in Wisconsin. Vehicle owners identify their vehicle make and model on registration materials,



and often we found that these identifiers did not match manufacturer specified models. As such, we discarded 325 (13% of original dataset) inconclusive vehicles from the data set since we could not confirm if these vehicles were indeed plug-in EVs. Due to these assumptions, the Wisconsin vehicle registrations are conservative compared to other sources [13]. Vehicles are binned within each of the five utility service territories based on registration zip code. While the majority of all PEV charging occurs overnight in owners' homes [22], this assumption may not accurately represent the service territory in which every vehicle is frequently charged. Vehicle registrations with the primary owner listed under non-Wisconsin zip codes were evenly distributed into all WI service territories.

Daily charging load curves are different for all EVs given vehicle characteristics like plug-in hybrid vs nonhybrid, size of the vehicle, and intended battery range. The PEV categories used here are plug-in hybrid electric vehicles with 20-mile (PHEV20) and 50-mile (PHEV50) battery ranges, plug-in hybrid sports utility vehicles with 20-mile (PHEV20 SUV) and 250-mile (BEV250 SUV) battery ranges, and battery-only electric vehicles with 100-mile (BEV100) and 250-mile (BEV100) battery ranges. All registered Wisconsin electric vehicle makes and models were binned into these six main categories, shown in Table 1.

**Table 1.** Number and percent share of registered Wisconsin PEVs by vehicle category [data from 12].

| Category | Counts | Percent |
|---|---|---|
| PHEV20 | 241 | 11.4% |
| PHEV50 | 980 | 46.2% |
| PHEV20 SUV | 101 | 4.8% |
| BEV100 | 405 | 19.1% |
| BEV250 | 305 | 14.4% |
| BEV250 SUV | 89 | 4.2% |

Daily charging load curves for each vehicle category were obtained from the National Renewable Energy Laboratory (NREL) report "Charging Electric Vehicles in Smart Cities: An EVI-Pro Analysis of Columbus Ohio" [23]. The EVI-Pro Analysis used by NREL to simulate charging profiles attempts to optimize vehicle charging with operating costs and assumes drivers have access to multiple charging stations throughout the day. The location of nearly 600 public charging stations in the State of Wisconsin [13] are shown in Figure 1b. The charging stations closely match the location of registered PEVs. EV charging load profiles from this study were used because of similar lifestyles and climatic conditions in Ohio and Wisconsin, the attention to differences in load profiles by vehicle type, load profiles which favor residential charging, and the exclusion of PEV-specific time-of-use pricing, which is not commonly used by Wisconsin utility companies [24]. We applied the daily charging load profiles of all six vehicle categories to current Wisconsin electric vehicles using a weighted average. A single charging profile curve was determined by weighting these six profiles with the current share of vehicles registered in Wisconsin as shown in Table 1. The six daily charging load curves from the NREL report and the Wisconsin weighted charging curve are shown in Figure 4. The single weighted load curve was applied to several Wisconsin PEV growth scenarios and assumed constant for every day of the year, ignoring different driving behaviors based on the day of the week and season.

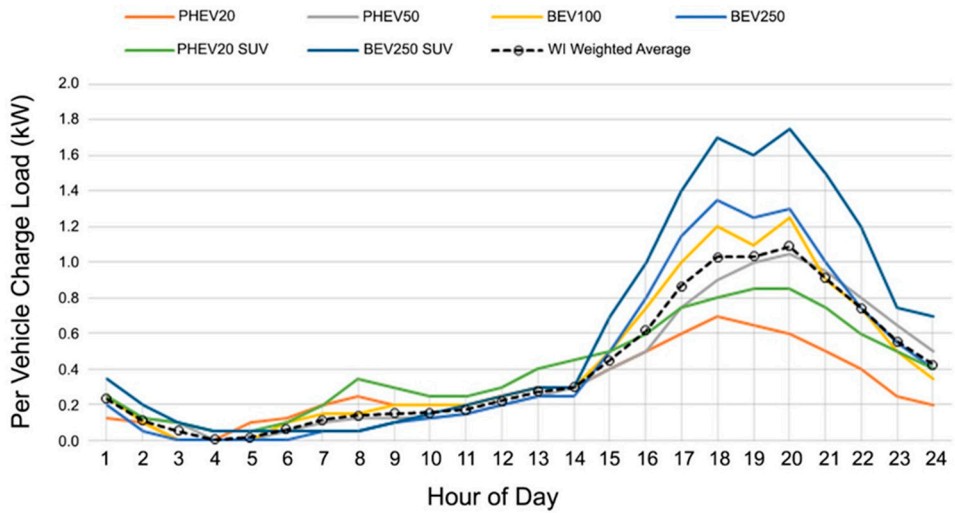

**Figure 4.** Daily charging load profiles for six vehicle categories, data from [23]. A single charging profile curve was determined by weighting these six profiles with the current share of vehicles registered in Wisconsin.

### 2.2. Growth Scenarios

Data from the 2019 Annual Energy Outlook (AEO2019) by the U.S. Energy Information Administration (EIA) was used to estimate PEV adoption in Wisconsin for the year 2030. The EIA recorded 255 million light-duty vehicles in the United States in 2018, and 0.43% of these were PEVs [7]. As of 2018, Wisconsin, with 2% of the U.S. population, has 0.19% of the total PEVs in the United States [12]. The AEO2019 report projects energy growth for many different industry sectors, with specific model assumptions for one reference case and six side cases. Under the reference case assumptions, the EIA projects that there will be 272.1 million light-duty vehicles (LDV) in the U.S. by 2030, with PEVs representing 5.1% of that national total. This EIA projection implies a compound annual growth rate of 0.54% per annum for the entire U.S. PEV stock.

There are three PEV growth scenarios used in this study: baseline, reference, and high-adoption. All projected vehicles are expected to utilize the electric grid for 100% of their charging needs. In many cases, PEV owners may also choose to install distributed generation resources, such as rooftop solar, which would offset their use of the electric grid for charging. Our assumption puts more strain on the grid than assuming some portion of future PEV owners will use distributed generation for vehicle charging.

The baseline scenario uses internal load growth assumptions of the ABB PROMOD software, described in the Section 2.3. These load growth assumptions use proprietary industry data accepted by transmission system planners. We did not add any new electric vehicle charging loads to the PROMOD load assumptions for the baseline scenario.

The reference growth scenario used here was determined using the EIA AEO2019 reference case assuming Wisconsin's adoption of PEVs continues to maintain a 0.19% share of PEVs in the U.S. in 2030. Wisconsin is assumed to share the same percentage of PEVs in 2030 and grow at the same rate as the U.S. on average. Given these assumptions, the reference growth scenario used in this study results in 26,600 electric vehicles registered in Wisconsin by 2030. Wisconsin currently has a 2.75% share of the U.S. light duty vehicle stock. Assuming Wisconsin maintains the same percentage of U.S. vehicles, the number of total vehicles for 2030 in Wisconsin is estimated to be 7,481,124. Assuming a light-duty vehicle stock of 7.48 million, PEVs would only make up 0.36% of all Wisconsin LDVs in 2030. The final, high-adoption growth scenario was selected assuming 5% of registered Wisconsin vehicles in 2030 will be plug-in electric vehicles, matching the 2030 national PEV penetration assumption of the AEO2019 [7]. This would result in over 373,000 PEVs in Wisconsin in 2030. This scenario would only be possible given the rapid adoption of electric vehicles by Wisconsin citizens.

The number of Wisconsin registered PEVs in 2018 and those estimated in 2030 for the reference and high-adoption growth scenarios are shown in Table 2. The 2030 load assumptions for the total state of Wisconsin for each of the three scenarios are shown in Table 3. While there is a two-orders-of-magnitude increase in Wisconsin plug-in electric vehicles between 2018 and 2030 under the high-adoption scenario, total energy load only increases less than a percentage as a result of these new vehicles.

**Table 2.** Number of PEVs by service territory in 2018 [data from 12] and the modeled number in 2030 under the reference and high-adoption growth scenarios. PEV increases over 2018 registrations are shown.

| | Utility | | | | | | Increase Over 2018 |
|---|---|---|---|---|---|---|---|
| | **MGE** | **NSP** | **WEPCO** | **WPL** | **WPS** | **Total** | |
| **2018 Registrations** | 355 | 272 | 1437 | 729 | 285 | 3077 | - |
| **Reference Growth** | 3072 | 2351 | 12,433 | 6309 | 2466 | 26,632 | 765% |
| **High Adoption** | 43,114 | 32,993 | 174,479 | 88,549 | 34,627 | 373,761 | 12046% |

**Table 3.** Modeled 2030 annual load (GWh/year) in Wisconsin utilities resulting from the baseline, reference, and high-adoption growth scenarios. Energy increase over baseline scenario are shown.

| | Utility | | | | | | Increase Over 2018 |
|---|---|---|---|---|---|---|---|
| | **MGE** | **NSP** | **WEPCO** | **WPL** | **WPS** | **Total** | |
| **Baseline Growth** | 4508 | 64,126 | 42,928 | 16,304 | 15,581 | 143,447 | - |
| **Reference Growth** | 4518 | 64,133 | 42,970 | 16,326 | 15,589 | 143,536 | 0.06% |
| **Progressive Growth** | 4650 | 64,191 | 43,483 | 16,591 | 15,734 | 144,649 | 0.84% |

*2.3. Electricity Market Simulation*

To simulate the regional transmission grid, we used PROMOD IV (ABB, Zurich, Switzerland, version 11.1) tool. PROMOD is an industry-standard software used by system planners today. MISO uses this tool for production cost simulation modeling in their annual transmission planning process [25]. PROMOD makes general assumptions about utility load and growth for projected years using both publicly and industry-proprietary available generation, load, and transmission system data. Notable assumptions made by the PROMOD model include announced generation additions and retirements; proposed transmission system additions; and simplified load profiles that differ only by weekdays, weekends, seasons, and holidays. The baseline Eastern Interconnect transmission system model was used, considering publicly known changes to which Wisconsin generators and transmission lines would likely be in operation by the year 2030.

A baseline load for 2030 is created by the simulator for each of the five Wisconsin service territories using these assumptions for each utility in 2030. We did not add any PEV charging load to the baseline scenario for 2030. To create the reference and high-growth load scenarios for each Wisconsin utility, the baseline load as modeled by PROMOD is increased considering the PEVs daily load charging curves and growth scenarios for each utility territory separately.

The built-in PROMOD load assumptions are considered a good projection for load growth. To confirm that the PROMOD load assumptions match practice, we compared historic PROMOD load assumptions with known Wisconsin load. The ABB PROMOD baseline load assumptions for Wisconsin in 2014 closely matched those calculated in the Strategic Energy Assessment (SEA) by the Public Service Commission of Wisconsin in the same year [26], with the exception that the Northern States Power utility territory in PROMOD does not disaggregate that load in Wisconsin from that in Minnesota. We kept the full NSP territory load—both Wisconsin and Minnesota—in this study.

## 3. Results

Box and whisker plots are used to comparatively visualize the differences in hourly LMPs of each utility service territory under each load growth scenario. These plots represent how the dataset is distributed for the year 2030 by summarizing data statistics. The modeled 2030 hourly LMPs and congestion components are shown in Figure 5, and the representative statistics are given in Tables 4 and 5. These results can be compared with the historic ex post LMPs used in MISO market settlements for each service territory, given in Figures 2 and 3. The line within the box represents the median, and the cross represents the average of all hourly LMP values in 2030. The third quantile (75 percentile) is presented from the median to the top of the box, and the first quantile (25 percentile) is represented from the median to the lower portion of the box. The whiskers above and below each box indicate the local maximum and minimum, excluding outliers which are more than 1.5 times the interquartile range.

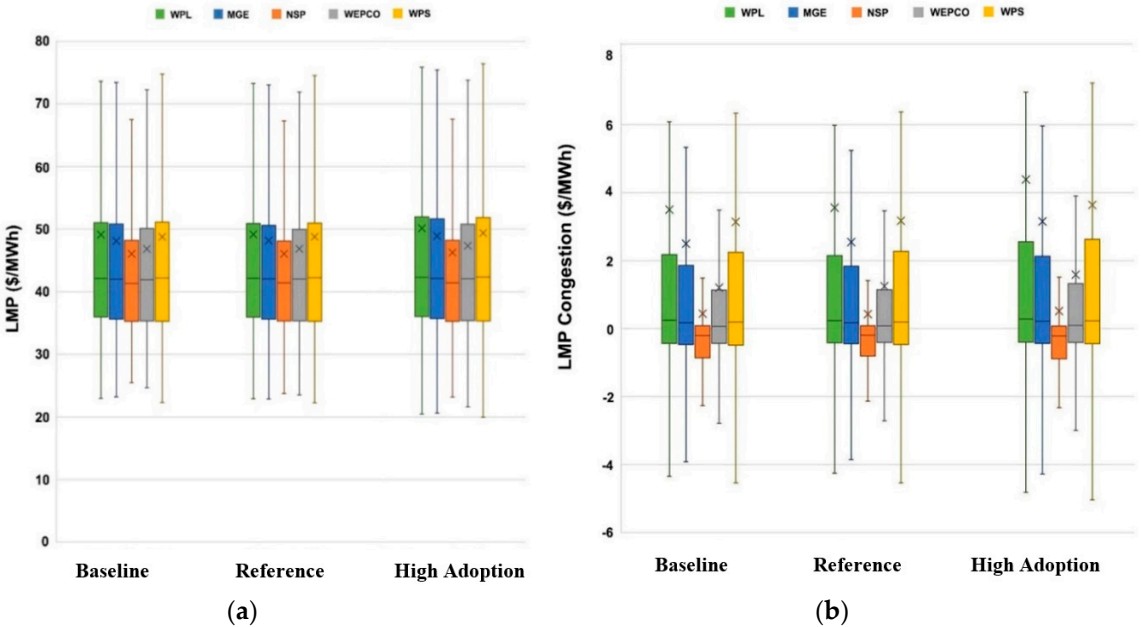

**Figure 5.** Total hourly LMPs (**a**) and congestion LMP component (**b**) on load buses of each Wisconsin service territory given all growth scenarios in 2030.

**Table 4.** Statistics of modeled hourly LMPs ($/MWh) for each utility in Wisconsin in 2030 under the three growth scenarios.

| | Utility | | | | |
|---|---|---|---|---|---|
| | MGE | NSP | WEPCO | WPL | WPS |
| | Baseline Scenario | | | | |
| Minimum | 8.36 | 8.54 | 9.91 | 8.35 | 8.68 |
| Lower Quartile (25%) | 35.63 | 35.26 | 35.32 | 35.95 | 35.26 |
| Median | 41.99 | 41.31 | 41.93 | 42.10 | 42.17 |
| Average | 48.11 | 46.05 | 46.82 | 49.11 | 48.75 |
| Upper Quartile (75%) | 50.80 | 48.18 | 50.12 | 51.03 | 51.14 |
| Maximum | 785.82 | 781.26 | 751.24 | 805.51 | 828.95 |
| | Reference Growth Scenario | | | | |
| Minimum | 8.03 | 8.00 | 8.27 | 8.03 | 8.10 |
| Lower Quartile (25%) | 35.61 | 35.30 | 35.33 | 35.93 | 35.26 |
| Median | 42.04 | 41.42 | 42.00 | 42.14 | 42.23 |
| Average | 48.16 | 46.04 | 46.86 | 49.17 | 48.78 |
| Upper Quartile (75%) | 50.59 | 48.10 | 49.96 | 50.88 | 50.97 |
| Maximum | 785.84 | 781.24 | 751.26 | 805.52 | 828.97 |
| | Progressive Growth Scenario | | | | |
| Minimum | 20.62 | 23.17 | 21.61 | 20.44 | 19.96 |
| Lower Quartile (25%) | 35.72 | 35.27 | 35.38 | 36.02 | 35.32 |
| Median | 42.10 | 41.43 | 42.06 | 42.30 | 42.34 |
| Average | 48.90 | 46.26 | 47.33 | 50.13 | 49.38 |
| Upper Quartile (75%) | 51.62 | 48.22 | 50.78 | 51.97 | 51.84 |
| Maximum | 764.31 | 760.68 | 726.78 | 786.13 | 812.32 |

**Table 5.** Statistics of the modeled congestion component of hourly LMPS ($/MWh) for each utility in Wisconsin in 2030 under the three growth scenarios.

| | Utility | | | | |
|---|---|---|---|---|---|
| | MGE | NSP | WEPCO | WPL | WPS |
| | Baseline Scenario | | | | |
| Minimum | −116.53 | −139.61 | −69.25 | −116.42 | −104.73 |
| Lower Quartile | −0.47 | −0.86 | −0.44 | −0.44 | −0.49 |
| Median | 0.17 | −0.21 | 0.06 | 0.24 | 0.19 |
| Average | 2.49 | 0.44 | 1.20 | 3.49 | 3.13 |
| Upper Quartile | 1.85 | 0.08 | 1.13 | 2.17 | 2.24 |
| Maximum | 212.02 | 200.70 | 212.11 | 283.80 | 278.96 |
| | Reference Growth Scenario | | | | |
| Minimum | −116.34 | −139.22 | −68.99 | −116.23 | −104.55 |
| Lower Quartile | −0.45 | −0.81 | −0.41 | −0.42 | −0.47 |
| Median | 0.17 | −0.20 | 0.08 | 0.23 | 0.19 |
| Average | 2.54 | 0.42 | 1.24 | 3.55 | 3.17 |
| Upper Quartile | 1.83 | 0.08 | 1.14 | 2.14 | 2.27 |
| Maximum | 210.05 | 202.03 | 209.68 | 281.93 | 275.03 |
| | Progressive Growth Scenario | | | | |
| Minimum | −115.98 | −138.47 | −68.41 | −115.87 | −104.21 |
| Lower Quartile | −0.44 | −0.89 | −0.41 | −0.40 | −0.45 |
| Median | 0.21 | −0.22 | 0.09 | 0.28 | 0.22 |
| Average | 3.15 | 0.51 | 1.58 | 4.38 | 3.63 |
| Upper Quartile | 2.12 | 0.07 | 1.32 | 2.55 | 2.62 |
| Maximum | 213.58 | 203.01 | 209.28 | 277.81 | 312.55 |

There is little change between the average hourly LMP values of each growth case compared to baseline, indicating little change in transmission congestion as a result of PEVs. The differences in average hourly LMPs from the baseline case is less than $1/MWh for all utilities, representing a change of less than 2%. The differences in average hourly LMP congestion from the baseline case is also less than $1/MW, but representing a higher change in congestion prices of 15–32% for all utilities in the high-growth scenario.

All utilities have LMPs with positive skewness, as seen in Figure 5. We note that while the interquartile range of LMPs increased between the baseline and high-adoption growth scenarios for all utilities, the maximum LMPs seen in the high-adoption case were less than those in the baseline case for all utilities (Table 4). The maximum congestion component of LMPs do not show this same behavior between the baseline and high-adoption cases (Table 5). A look at the load and LMP curves on the day in which the maximum LMP occurred reveal that the afternoon shifting of load due to EV charging also results in a shift and slight flattening of energy prices on that day. This behavior is demonstrated in Figure 6, which shows the hourly LMP and load curves for Wisconsin Power and Light utility on 8–10 July 2030. The maximum LMP was recorded on 9 July for all utilities in all scenarios.

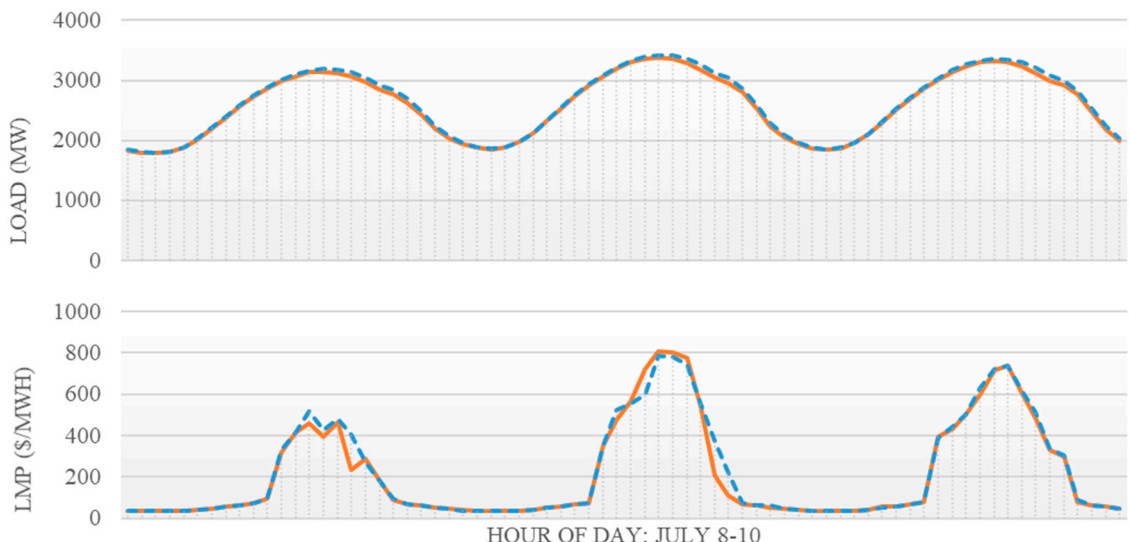

**Figure 6.** Simulated load (top) and LMP (bottom) curves for Wisconsin Power and Light from 8–10 July 2030 for the baseline (solid line) and high-adoption (dashed line) scenarios. The maximum LMP of the year in either scenario occurs on 9 July 2030. Load shifts to the afternoon in the high-adoption scenario.

## 4. Policy Discussion

The policy implications of these results are mixed. The relatively small increases in electricity prices resulting from modeled electric vehicle adoption in Wisconsin do not indicate that transmission system upgrades will be needed in direct response to the growth in charging load. On the other hand, the increase in congestion could be mitigated by aligning electric vehicle charging schedules with wholesale electricity prices. By shifting charging load to times of traditionally low demand—such as at night—the daily dispersion of LMPs is expected to decrease [27]. This work showed evidence of marginal price flattening due to controlled load shifting in the high-adoption scenario. Load shifting can also help during emergency generation events [28], which occur during extreme weather events as seen in Midwest winters.

This work says nothing about upgrades that may be necessary on the distribution system, owned and operated by utility companies. Many distribution utility companies are adopting retail rate structures to encourage their customers to use electricity during off-peak times when wholesale electricity prices are low [29], and the same practice could be useful in mitigating distribution system upgrades. Time-of-use pricing is the most common retail rate structure to encourage demand shifting

for all residential customers, but many utilities are implementing similar structures specifically for residential EV charging outlets to shift drivers' charging behaviors [30]. The temporal resolution of off-peak rate structures could be more granular by exposing electric vehicles to real-time wholesale electricity price signals and using electric vehicle charging as a demand response asset to smooth electricity prices [31].

As Wisconsinites continue to adopt electric vehicles, there will be more opportunity to shift load to times of historically low demand if a need does arise on either the distribution or the transmission system. Implementing rate structures that align with off-peak load is the choice of distribution utility companies, while encouraging the use of electric vehicle charging as a demand response asset would require appropriate State legislation, regulations, and business incentives.

## 5. Conclusions

This analysis makes many assumptions about load, generation, and transmission growth in the state of Wisconsin, relying heavily on the baseline assumptions of the ABB PROMOD modeling software. As such, only comparative results between the baseline case and each tested case are robust to the levels of the assumptions used.

There are minimal changes in the utility territory electricity prices between the AEO reference and the baseline growth scenarios. There is a less than $0.50/MWh (1%) change in LMPs and less than $0.10/MWh (4%) change in congestion LMP for these cases.

The high-adoption case showed a greater change in the congestion LMP prices compared to baseline, but still little difference in overall LMPs for all service territories. Average LMPs changed by less $1.05/MWh (2%) for all service territories. This is less than the historic annual changes of average wholesale prices seen in MISO due to normal operation of the grid, as seen in Figures 2 and 3. LMP congestion also changed by less than $1.00/MWh for the high-adoption case, but this represents a higher percentage change between 16% and 32% of baseline congestion prices.

Overall, electric vehicle growth to 5% of all light-duty vehicles in Wisconsin by 2030 would not result in significant changes to the LMPs of the five Wisconsin service territories considered given model assumptions. The increase in LMPs resulting from electric vehicle adoption is expected to be less than that seen due to usual changes in the electric grid on an annual basis—e.g., transmission system upgrades, generator interconnections and retirements, conventional load growth, and extreme weather events—based on historic electricity prices seen in Figures 2 and 3. There are, however, moderate relative increases in congestion pricing, an indication of increased power flow congestion on the transmission grid.

This study considers how unmanaged electric vehicle charging growth will likely impact the existing Wisconsin transmission grid. There will be unforeseen electric grid system upgrades that may lessen the impact of electric vehicle load, including owners' decisions to install residential distributed generation resources. In a high-growth scenario, it is likely that distribution utilities would make system upgrades and incentivize customers to charge at desired times of day to specifically accommodate the increase in charging load. For these reasons, the wholesale pricing increases seen at 5% PEVs in this study may be an overestimate. It is possible that increased congestion pricing due to electric vehicle growth may not been seen until more than 5% of Wisconsinites' vehicles are charging from the grid.

This work is the result of a collaboration between state government and the university, which is unfortunately rare within the academic literature. The study is motivated by concerns held by the Public Service Commission of Wisconsin, the state electric grid regulatory body. The regulator is most concerned with how modest load-growth scenarios would impact state electricity prices, without assuming the implementation of new policies. This work exposes the questions of importance to regulators and transmission planners in anticipating changes in wholesale congestion pricing.

## 6. Limitations

The Wisconsin vehicle registration data was made available by the Wisconsin Department of Transportation on a one-time basis. The authors do not have access to more recent vehicle registration data. We recognize the gap in time between collecting data, analysis, and publishing results as a trait of academia, and regret that the registration data used is a year older than the publication date.

The PEV growth scenarios used in this study were chosen in consultation with the PSCW to answer questions they posed to the researchers. Similarly, researchers were asked to use the modeling tools available to Commission engineering staff and not academic-built modeling tools. The use of PSCW tools enabled reputable results that Commission staff can easily interpret and trust. In a purely academic study, more aggressive load growth scenarios, modeling assumptions that include intelligent charging and coupled distributed generation–PEV charging control schema, and publicly reviewed production cost modeling tools may have been used.

**Author Contributions:** Conceptualization, M.Z., A.B., and G.N.; Methodology, M.Z. and A.B.; Data curation, M.Z. and A.B.; Formal analysis, M.Z. and A.B.; Project administration, M.Z.; Software, A.B.; Visualization, M.Z.; Writing—original draft, M.Z., A.B., and G.N.; Writing—review and editing, M.Z., A.B., and G.N.; Funding acquisition, M.Z. and A.B. All authors have read and agreed to the published version of the manuscript.

**Funding:** Funding for this work came in the form of paid employee time for authors M.Z. and A.B. from the Public Service Commission of Wisconsin (PSCW). Both M.Z. and A.B. were graduate students at the University of Wisconsin—Madison and worked part time for the PSCW during the completion of this study. A.B.'s academic National Science Foundation fellowship funding was not used to support this work.

**Acknowledgments:** The authors acknowledge and thank the Public Service Commission of Wisconsin for their partial support of this work. The conclusions and views expressed here are the authors' alone and not those of the PSCW.

**Conflicts of Interest:** The authors declare no conflicts of interest as they pertain to funding for this work. Raw data used for modeling is confidential and protected by Nondisclosure Agreements under the State of Wisconsin. Results of this study were approved for publication by PSCW senior staff member Randel Pilo.

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
