# Peer review of "The Impacts of Electric Vehicle Growth on Wholesale Electricity Prices in Wisconsin"

_wevj, doi:10.3390/wevj11020032_

Round 1

Reviewer 1 Report

This study aims at discussing the impact on transmission congestion using wholesale electricity pricing as a proxy for system impacts, based on the projected electric vehicle growth in the State of Wisconsin.

Although there is a gap in the literature, for such specific study, i feel that the study (in its present state) needs a lot of improvements to fill that gap. 

1) The literature review / introduction, is too short and dosen't address all the aspects of concepts in this study. Some specific point regarding the same are enlisted below:

  • Line 38-40 : You talk about adoption rates, but the example that you have given is about the estimate of the number of cars in 2040. Reading which as a reader i don't get the information about the adoption rates. Either give example of the "rate" or amend the statements to match the example.
  • Line 46: Estimate is from September 2018, isn't a more recent estimate available ? There would be a significant difference between the number of electric cars then and now in 2020.
  • Overall the section 1.1. is too brief. More details need to be added to it. Especially the differences between BEV and PEV should be explained, or it should be mentioned clearly that BEV and PHEV are sub categories of PEV. Like it is done in one of the papers published in this journal itself. file:///C:/Users/Administrator/Desktop/New%20folder/wevj-08-00789.pdf Moreover, the figure 1a is from 2018 and 1b is from 2019. It would be more appropriate to take the distributions from the same time.
  • In section 1.2, it would be good to add a few lines about Smart Electricity Meters which enable consumers to get real time information about energy prices and consumption. Although these are for the retail market, but it would highlight that even with current technology, consumer don't have access to whole sale market prices.
  • From a reader's point of view, in the introduction, mentioning information about the structure of the article and particular aims of the study would add value. The objective is stated clearly, which is good.
  • Overall Introduction should be detailed more.

2) For materials and methods, there are lot of assumptions made in the study, which has also been stated in the conclusions. But the theoretical basis of some of the assumptions are missing. 

  • Lines 114-115: What was inconclusive, what was the basis of discarding the entries, how many entries were discarded. This has to be explained in detail.
  • Line 133-138 - After looking at the NREL report (reference 14), i feel that the manusricpt needs to justify using the load profiles mentioned in that study. The study has taken into consideration the commercial vehicle charging as well. In the manusricpt there was no mention of the number of commercial charging stations in the targeted region. The results which point to the charging of cars in the afternoon could well be related to the commercial charging station available in a different location as compared to the residential address. Which in my opinion creates doubts about the reliability of the results.
  • The growth scenarios considered, also need to be re-thought in my opinion to give more relatable results.
  • Line 160-168 - The baseline scenario discussed dosen't mention the number of PEV's it just takes into consideration the load. It refers to explaination in the next section, but i am not sure which one. Exact reference should be put. Secondly for the reference scenario, 0.19% growth rate is not suitable. The rate of adoption should be compounded the inevitable growth of the PEVs, suggested in so many studies in the literature should be considered. This has been done for high adoption when the predicted national adoption rate (it should considered as 5.1% not 5%, mention only the EIA figure or AEO2019 figure) has been used. Also, the figure of 7.48 million in line 167, need to have a basis which should be explained.
  • In the simulation, the authors need to explain what developments (such as steps by the energy suppliers to stabilize the grid, add capacity etc)  does the simulation take into consideration or it takes no developments into account. If it takes the developments in to account, then it would account for the low rise in congestion even in case of high adoption rate. If it dosen't take into account the development scenarios, then the simulation results cannot be treated correct, as assuming that there will be no development in the next 10 years is not suitable. 
  • The authors also need to take into account the fact that some of the PEV users might be prosumers, who would use their own generated electricity for charging the PEV's. This could be one of the reasons that even in high adoption rate, the impact on grid or prices wouldn't be that much. 
  • The policy discussion and conclusions need to address all these issues as well.

Overall the theme of the paper is interesting but i feel that it needs major improvements, for it to be considered for publication. 

Author Response

Response to reviewer comments

Reviewer 1

Open Review

English language and style

( ) Extensive editing of English language and style required 
(x) Moderate English changes required 
( ) English language and style are fine/minor spell check required 
( ) I don't feel qualified to judge about the English language and style 

Yes

Can be improved

Must be improved

Not applicable

Does the introduction provide sufficient background and include all relevant references?

( )

( )

(x)

( )

Is the research design appropriate?

( )

( )

(x)

( )

Are the methods adequately described?

( )

(x)

( )

( )

Are the results clearly presented?

( )

( )

(x)

( )

Are the conclusions supported by the results?

( )

( )

(x)

( )

Comments and Suggestions for Authors

This study aims at discussing the impact on transmission congestion using wholesale electricity pricing as a proxy for system impacts, based on the projected electric vehicle growth in the State of Wisconsin.

Although there is a gap in the literature, for such specific study, i feel that the study (in its present state) needs a lot of improvements to fill that gap. 

1) The literature review / introduction, is too short and dosen't address all the aspects of concepts in this study. Some specific point regarding the same are enlisted below:

  • Line 38-40: You talk about adoption rates, but the example that you have given is about the estimate of the number of cars in 2040. Reading which as a reader i don't get the information about the adoption rates. Either give example of the "rate" or amend the statements to match the example.

We have removed the introductory statement on adoption rates and have revised the introduction (now page 2, lines 72-73) to read: there are several organizations with competing future projection estimates of PEV adoption. This sentence will be used as a transition from current EV estimates to future projections and clarify the statements we are making regarding adoption estimates in the US.

  • Line 46:Estimate is from September 2018, isn't a more recent estimate available ? There would be a significant difference between the number of electric cars then and now in 2020.

This proprietary data was made available by the Wisconsin Department of Transportation on a one-time basis. We do not have access to more updated registration numbers. We recognize the gap in time between collecting data, analysis and publishing results as a trait of academia.

  • Overall the section 1.1. is too brief. More details need to be added to it. Especially the differences between BEV and PEV should be explained, or it should be mentioned clearly that BEV and PHEV are sub categories of PEV. Like it is done in one of the papers published in this journal itself. file:///C:/Users/Administrator/Desktop/New%20folder/wevj-08-00789.pdf Moreover, the figure 1a is from 2018 and 1b is from 2019. It would be more appropriate to take the distributions from the same time.

We have added more detail to Section 1.1, most notably the addition of two paragraphs (page 2, line 79 – page 3, line 102) and a map of all public charging stations in Wisconsin (Fig 1b). The utility jurisdiction boundaries were unchanged between years 2018 and 2019 and the map is relevant for both years. The added paragraphs read:

The transportation technologies discussed will be mainly passenger vehicles, light-duty vehicles, and fleet vehicles registered with the Wisconsin Department of Transportation (WisDOT). There are three main categories of PEVs: hybrid-electric vehicles (HEVs), plug-in hybrid electric vehicles (PHEVs), and battery electric vehicles (BEVs). HEVs use an internal combustion engine (ICE) system along with an electric propulsion system, resulting in improved fuel economy and performance compared to standard ICE vehicles. These improvements occur through regenerative braking, idling, and electric-only drive. HEVs also use gasoline or diesel as fuel and do not require battery charging. PHEVs use an ICE system along with an electric propulsion system and a rechargeable battery. These batteries are charged through an outlet and allow PHEVs to drive extended distances on an electric charge. These vehicles typically run on only electricity until the battery depletes, at which point the ICE engages and then powers the vehicle. BEVs use a rechargeable battery and run on electricity alone. BEVs are charged by plugging into an outlet and can travel between 100 and 335 miles on a single charge, depending on the battery size, make, and model. In this report, plug-in electric vehicles are those vehicles that utilize battery charging: PHEVs and BEVs. HEVs do not use electricity from the grid, therefore they are not applicable to the transmission impacts studied here.

In September 2018, there were just over 2,100 plug-in electric vehicles registered with the Wisconsin Department of Transportation [12]. In February 2019, the Alternative Fuels Data Center identified nearly 600 public charging outlets in the State of Wisconsin of all levels [13]. The geographical dispersion of registered electric vehicles and public charging stations are shown in Figures 1(a) and (b), respectively. Figure 1(c) shows the distribution utility jurisdictions in Wisconsin [30]. Unsurprisingly the location of public charging stations closely aligns with the locations of registered plug-in electric vehicles. Level 1 chargers are standard 110V outlets. Level 2 chargers are 240V chargers like those used in the United States for clothes washing machines. Level 3 chargers are fast DC chargers used only by the long-range BEVs [1].

  • In section 1.2, it would be good to add a few lines about Smart Electricity Meters which enable consumers to get real time information about energy prices and consumption. Although these are for the retail market, but it would highlight that even with current technology, consumer don't have access to whole sale market prices.

We worry that a discussion on smart meters and retail pricing could distract from the conversation about wholesale pricing. We have revised page 4, lines 139-141 to read: Retail customers are not typically exposed to the real-time price volatility of wholesale electricity prices, though advances in smart metering technology has renewed a debate that they should be.

  • From a reader's point of view, in the introduction,mentioning information about the structure of the article and particular aims of the study would add value. The objective is stated clearly, which is good.

We included the following paragraph (page 2, lines 60-68) to the introduction which provides the article structure:

The structure of this article is as follows. A review of the current electric vehicles registered in the State of Wisconsin is given in Section 1.1. An overview of the regional electric grid and the many system operators in the State is given in Section 1.2 A discussion of the differences between wholesale and retail electricity pricing, including the regulatory management of prices, is also included in Section 1.2. Section 2 reviews the methods used in this study, namely estimates of electric vehicle growth, EV charging profiles, and the electricity market simulation tool used. Section 3 describes the wholesale electricity price results for the three electric vehicle growth scenarios. A discussion of the policy implications of the change in wholesale pricing due to EV growth is given in Section 4. Conclusions are presented in Section 5.

  • Overall Introduction should be detailed more.

We have rewritten all the introduction section, including the aforementioned changes to Sections 1.1 and 1.2. We believe this introduction better explains the challenges faced by transmission owners and regulatory bodies regarding transmission planning and anticipating changes in wholesale congestion pricing. The new introduction (prior to Section 1.1) reads:

The transmission system was built to accommodate delivery of power from generators to loads given a snapshot in time. As both the world’s generation mix and load profiles evolve, electric transmission systems must similarly adapt. One indication that new transmission lines are needed is an increase in power flow congestion on existing lines. A concern over how the transmission grid must evolve to accommodate increased electric vehicle charging load has arisen among practitioners.

Transmission owners are tasked with maintaining a reliable electric transmission system and installing transmission infrastructure when necessary. State regulators are tasked with protecting ratepayer interests and ensuring any increase in electricity operational costs are justified by system need. The delicate balance of transmission system planning given changing generation and load is overseen by the Regional Transmission Operators, of which both transmission owners and regulators are participatory stakeholders. In order to plan necessary upgrades to the electric grid, practitioners first test how anticipated changes would strain the existing electric grid. Only then will operators consider what novel control schemes, metering tools, and grid modernization efforts common in the research literature could be applied to address these concerns.

This research was motivated to answer the question: Will the projected growth in Wisconsin plug-in electric vehicles (PEV) increase electric system congestion and require the construction of new transmission lines? To answer this question, we use the grid modeling tools available to operators, assume no changes to the existing Wisconsin electric grid, and use realistic electric vehicle growth assumptions. This research required access to proprietary electric grid data, not commonly available to researchers for security purposes. This work is written to address the concerns of transmission system practitioners, upon which the research community can build.

Current electric vehicle research considers technology adoption projections and the infrastructure needed to support PEV growth [1-3], and a smaller subset of studies have considered the specific impacts on distribution and transmission electric grids [4-6]. There are few studies specific to the Midwestern United States [3], but no existing work for Wisconsin regarding PEVs and transmission congestion. This study analyzes how electrification of light-duty vehicles will transform the use of the electric power grid along bulk transmission lines. This work considers how projected electric vehicle growth in the State of Wisconsin would impact transmission congestion, using wholesale electricity pricing as a proxy for system impacts. The objective of this research is to create a better understanding of how an emerging PEV market can impact transmission system need.

2) For materials and methods, there are lot of assumptions made in the study, which has also been stated in the conclusions. But the theoretical basis of some of the assumptions are missing. 

  • Lines 114-115: What was inconclusive, what was the basis of discarding the entries, how many entries were discarded. This has to be explained in detail.

In section 2.1, we revised the statement regarding the inconclusive data to include the total number of inconclusive entries and the reasoning for discarding the entries. Page 5, lines 174-192 now read:

Vehicle owners identify their vehicle make and model on registration materials, and often we found that these identifiers did not match manufacturer specified models. As such, we discarded 325 (13% of original dataset) inconclusive vehicles from the data set since we could not confirm if these vehicles were indeed plug-in EVs. Due to these assumptions, the Wisconsin vehicle registrations are conservative compared to other sources [13]. Vehicles are binned within each of the five utility service territories based on registration zip code. While the majority of all PEV charging occurs overnight in owners’ homes [21], this assumption may not accurately represent the service territory in which every vehicle is frequently charged. Vehicle registrations with the primary owner listed under non-Wisconsin zip codes were evenly distributed into all WI service territories.

  • Line 133-138- After looking at the NREL report (reference 14), i feel that the manusricpt needs to justify using the load profiles mentioned in that study. The study has taken into consideration the commercial vehicle charging as well. In the manusricpt there was no mention of the number of commercial charging stations in the targeted region. The results which point to the charging of cars in the afternoon could well be related to the commercial charging station available in a different location as compared to the residential address. Which in my opinion creates doubts about the reliability of the results.

Thank you for pointing this out. We have added Figure 1b which shows the volume and location of public charging stations in Wisconsin. We have additionally added the following to Section 2.1 (page 6, lines 198-201) to discuss the charging profile assumptions in the report and how this relates to Wisconsin. The added language reads:

The EVI-Pro Analysis used by NREL to simulate charging profiles attempts to optimize vehicle charging with operating costs and assumes drivers have access to multiple charging stations throughout the day. The location of nearly 600 public charging stations in the State of Wisconsin [13] are shown in Figure 1(b). The charging stations closely match the location of registered PEVs.

  • The growth scenarios considered, also need to be re-thought in my opinion to give more relatable results.

These growth scenarios were chosen in consultation with the Public Service Commission of Wisconsin to answer questions they posed to the researchers. Like the Department of Transportation data, we do not have the authority to choose different growth scenarios. We hope the editor and reviewer will understand this constraint. We believe this work has value in exposing what questions are of importance to regulators and transmission planners, which we tried to better address in the introductory paragraphs.

  • Line 160-168 - The baseline scenario discussed dosen't mention the number of PEV's it just takes into consideration the load. It refers to explaination in the next section, but i am not sure which one. Exact reference should be put. Secondly for the reference scenario, 0.19% growth rate is not suitable. The rate of adoption should be compounded the inevitable growth of the PEVs, suggested in so many studies in the literature should be considered. This has been done for high adoption when the predicted national adoption rate (it should considered as 5.1% not 5%, mention only the EIA figure or AEO2019 figure) has been used. Also, the figure of 7.48 million in line 167, need to have a basis which should be explained.

Section 2.2 has been revised to discuss the PEV growth scenarios and the vehicle estimate calculations. We hope that this is more understandable. Page 7, lines 232-243 now read:

The baseline scenario uses internal load growth assumptions of the ABB PROMOD software, described in the Section 2.3. These load growth assumptions use proprietary industry data accepted by transmission system planners. We did not add any new electric vehicle charging loads to the PROMOD load assumptions for the baseline scenario.

The reference growth scenario used here was determined using the EIA AEO2019 reference case assuming Wisconsin’s adoption of PEVs continues to maintain a 0.19% share of PEVs in the U.S. in 2030. Wisconsin is assumed to share the same percentage of PEVs in 2030 and grow at the same rate as the U.S. on average. Given these assumptions, the reference growth scenario used in this study results in 26,600 electric vehicles registered in Wisconsin by 2030. Wisconsin currently has a 2.75% share of the U.S. light duty vehicle stock. Assuming Wisconsin maintains the same percentage of U.S. vehicles, total vehicles for 2030 in Wisconsin is estimated to be 7,481,124. Assuming a light-duty vehicle stock of 7.48 million, PEVs would only make up 0.36% of all Wisconsin LDVs in 2030.

  • In the simulation, the authors need to explain what developments (such as steps by the energy suppliers to stabilize the grid, add capacity etc)  does the simulation take into consideration or it takes no developments into account. If it takes the developments in to account, then it would account for the low rise in congestion even in case of high adoption rate. If it dosen't take into account the development scenarios, then the simulation results cannot be treated correct, as assuming that there will be no development in the next 10 years is not suitable. 

Thank you for this comment. The transmission upgrades considered in this study are those announced by all utility and transmission owners in the MISO region. We have rewritten Section 2.3 to better describe the transmission planning assumptions. This section (page 8, lines 261-282) now reads:

To simulate the regional transmission grid, we used ABB’s PROMOD IV (version 11.1) tool. PROMOD is an industry-standard software used by system planners today. MISO uses this tool for production cost simulation modeling in their annual transmission planning process [23]. PROMOD makes general assumptions about utility load and growth for projected years using both publicly and industry-proprietary available generation, load and transmission system data. Notable assumptions made by the PROMOD model include announced generation additions and retirements, proposed transmission system additions, and simplified load profiles that differ only by weekdays, weekends, seasons, and holidays. The baseline Eastern Interconnect transmission system model was used, considering publicly known changes to which Wisconsin generators and transmission lines would likely be in operation by the year 2030.

A baseline load for 2030 is created by the simulator for each of the five Wisconsin service territories using these assumptions for each utility in 2030. We did not add any PEV charging load to the baseline scenario for 2030. To create the reference and high growth load scenarios for each Wisconsin utility, the baseline load as modeled by PROMOD is increased considering the PEVs daily load charging curves and growth scenarios for each utility territory separately.

The built-in PROMOD load assumptions are considered a good projection for load growth. To confirm that the PROMOD load assumptions match practice, we compared historic PROMOD load assumptions with known Wisconsin load. The ABB PROMOD baseline load assumptions for Wisconsin in 2014 closely matched those calculated in the Strategic Energy Assessment (SEA) by the Public Service Commission of Wisconsin in the same year [24], with the exception that the Northern States Power utility territory in PROMOD does not disaggregate that load in Wisconsin from that in Minnesota. We kept the full NSP territory load – both Wisconsin and Minnesota – in this study.

  • The authors also need to take into account the fact that some of the PEV users might be prosumers, who would use their own generated electricity for charging the PEV's. This could be one of the reasons that even in high adoption rate, the impact on grid or prices wouldn't be that much. 

Anticipated changes in distributed generation in 2030 are included in the utility load projections of the baseline scenario. We assumed all additional PEV in the reference and high adoption scenarios will utilize the electrical grid rather than using personal electricity. This assumption is briefly noted in the aforementioned changes to Section 2.3 and in the newly revised Section 2.2. Section 2.2, page 7 lines 226-231 reads:

There are three PEV growth scenarios used in this study: baseline, reference, and high adoption. All projected vehicles are expected to utilize the electric grid for 100% of their charging needs. In many cases, PEV owners may also choose to install distributed generation resources, such as rooftop solar, which would offset their use of the electric grid for charging. Our assumption puts more strain on the grid than assuming some portion of future PEV owners will use distributed generation for vehicle charging.

  • The policy discussion and conclusions need to address all these issues as well.

Additional assumptions were added to the conclusion section. Page 15, lines 370-378 now read:

This study considers how unmanaged electric vehicle charging growth would impact the existing Wisconsin transmission grid. There will be unforeseen electric grid system upgrades that may lessen the impact of electric vehicle load, including owners’ decisions to install residential distributed generation resources. In a high growth scenario, it is likely that distribution utilities would make system upgrades and incentivize customers to charge at desired times of day to specifically accommodate the increase in charging load. For these reasons, the wholesale pricing increases seen at 5% PEVs in this study may be an overestimate. It is possible that increased congestion pricing due to electric vehicle growth may not been seen until more than 5% of Wisconsinites’ vehicles are charging from the grid

Overall the theme of the paper is interesting but i feel that it needs major improvements, for it to be considered for publication. 

Thank you for taking the time to review this work so thoroughly. We hope that we have addressed all your concerns.

Reviewer 2 Report

In general the paper discusses the impact of EV on a specific market. Although there is nothing new in the paper but the information are worthwhile, however, i wish the authors to discuss the results in more detail as the significance of such papers should be in results and discussion 

Author Response

Reviewer 1

Open Review

English language and style

( ) Extensive editing of English language and style required 
(x) Moderate English changes required 
( ) English language and style are fine/minor spell check required 
( ) I don't feel qualified to judge about the English language and style 

Yes

Can be improved

Must be improved

Not applicable

Does the introduction provide sufficient background and include all relevant references?

( )

( )

(x)

( )

Is the research design appropriate?

( )

( )

(x)

( )

Are the methods adequately described?

( )

(x)

( )

( )

Are the results clearly presented?

( )

( )

(x)

( )

Are the conclusions supported by the results?

( )

( )

(x)

( )

Comments and Suggestions for Authors

This study aims at discussing the impact on transmission congestion using wholesale electricity pricing as a proxy for system impacts, based on the projected electric vehicle growth in the State of Wisconsin.

Although there is a gap in the literature, for such specific study, i feel that the study (in its present state) needs a lot of improvements to fill that gap. 

1) The literature review / introduction, is too short and dosen't address all the aspects of concepts in this study. Some specific point regarding the same are enlisted below:

  • Line 38-40: You talk about adoption rates, but the example that you have given is about the estimate of the number of cars in 2040. Reading which as a reader i don't get the information about the adoption rates. Either give example of the "rate" or amend the statements to match the example.

We have removed the introductory statement on adoption rates and have revised the introduction (now page 2, lines 72-73) to read: there are several organizations with competing future projection estimates of PEV adoption. This sentence will be used as a transition from current EV estimates to future projections and clarify the statements we are making regarding adoption estimates in the US.

  • Line 46:Estimate is from September 2018, isn't a more recent estimate available ? There would be a significant difference between the number of electric cars then and now in 2020.

This proprietary data was made available by the Wisconsin Department of Transportation on a one-time basis. We do not have access to more updated registration numbers. We recognize the gap in time between collecting data, analysis and publishing results as a trait of academia.

  • Overall the section 1.1. is too brief. More details need to be added to it. Especially the differences between BEV and PEV should be explained, or it should be mentioned clearly that BEV and PHEV are sub categories of PEV. Like it is done in one of the papers published in this journal itself. file:///C:/Users/Administrator/Desktop/New%20folder/wevj-08-00789.pdf Moreover, the figure 1a is from 2018 and 1b is from 2019. It would be more appropriate to take the distributions from the same time.

We have added more detail to Section 1.1, most notably the addition of two paragraphs (page 2, line 79 – page 3, line 102) and a map of all public charging stations in Wisconsin (Fig 1b). The utility jurisdiction boundaries were unchanged between years 2018 and 2019 and the map is relevant for both years. The added paragraphs read:

The transportation technologies discussed will be mainly passenger vehicles, light-duty vehicles, and fleet vehicles registered with the Wisconsin Department of Transportation (WisDOT). There are three main categories of PEVs: hybrid-electric vehicles (HEVs), plug-in hybrid electric vehicles (PHEVs), and battery electric vehicles (BEVs). HEVs use an internal combustion engine (ICE) system along with an electric propulsion system, resulting in improved fuel economy and performance compared to standard ICE vehicles. These improvements occur through regenerative braking, idling, and electric-only drive. HEVs also use gasoline or diesel as fuel and do not require battery charging. PHEVs use an ICE system along with an electric propulsion system and a rechargeable battery. These batteries are charged through an outlet and allow PHEVs to drive extended distances on an electric charge. These vehicles typically run on only electricity until the battery depletes, at which point the ICE engages and then powers the vehicle. BEVs use a rechargeable battery and run on electricity alone. BEVs are charged by plugging into an outlet and can travel between 100 and 335 miles on a single charge, depending on the battery size, make, and model. In this report, plug-in electric vehicles are those vehicles that utilize battery charging: PHEVs and BEVs. HEVs do not use electricity from the grid, therefore they are not applicable to the transmission impacts studied here.

In September 2018, there were just over 2,100 plug-in electric vehicles registered with the Wisconsin Department of Transportation [12]. In February 2019, the Alternative Fuels Data Center identified nearly 600 public charging outlets in the State of Wisconsin of all levels [13]. The geographical dispersion of registered electric vehicles and public charging stations are shown in Figures 1(a) and (b), respectively. Figure 1(c) shows the distribution utility jurisdictions in Wisconsin [30]. Unsurprisingly the location of public charging stations closely aligns with the locations of registered plug-in electric vehicles. Level 1 chargers are standard 110V outlets. Level 2 chargers are 240V chargers like those used in the United States for clothes washing machines. Level 3 chargers are fast DC chargers used only by the long-range BEVs [1].

  • In section 1.2, it would be good to add a few lines about Smart Electricity Meters which enable consumers to get real time information about energy prices and consumption. Although these are for the retail market, but it would highlight that even with current technology, consumer don't have access to whole sale market prices.

We worry that a discussion on smart meters and retail pricing could distract from the conversation about wholesale pricing. We have revised page 4, lines 139-141 to read: Retail customers are not typically exposed to the real-time price volatility of wholesale electricity prices, though advances in smart metering technology has renewed a debate that they should be.

  • From a reader's point of view, in the introduction,mentioning information about the structure of the article and particular aims of the study would add value. The objective is stated clearly, which is good.

We included the following paragraph (page 2, lines 60-68) to the introduction which provides the article structure:

The structure of this article is as follows. A review of the current electric vehicles registered in the State of Wisconsin is given in Section 1.1. An overview of the regional electric grid and the many system operators in the State is given in Section 1.2 A discussion of the differences between wholesale and retail electricity pricing, including the regulatory management of prices, is also included in Section 1.2. Section 2 reviews the methods used in this study, namely estimates of electric vehicle growth, EV charging profiles, and the electricity market simulation tool used. Section 3 describes the wholesale electricity price results for the three electric vehicle growth scenarios. A discussion of the policy implications of the change in wholesale pricing due to EV growth is given in Section 4. Conclusions are presented in Section 5.

  • Overall Introduction should be detailed more.

We have rewritten all the introduction section, including the aforementioned changes to Sections 1.1 and 1.2. We believe this introduction better explains the challenges faced by transmission owners and regulatory bodies regarding transmission planning and anticipating changes in wholesale congestion pricing. The new introduction (prior to Section 1.1) reads:

The transmission system was built to accommodate delivery of power from generators to loads given a snapshot in time. As both the world’s generation mix and load profiles evolve, electric transmission systems must similarly adapt. One indication that new transmission lines are needed is an increase in power flow congestion on existing lines. A concern over how the transmission grid must evolve to accommodate increased electric vehicle charging load has arisen among practitioners.

Transmission owners are tasked with maintaining a reliable electric transmission system and installing transmission infrastructure when necessary. State regulators are tasked with protecting ratepayer interests and ensuring any increase in electricity operational costs are justified by system need. The delicate balance of transmission system planning given changing generation and load is overseen by the Regional Transmission Operators, of which both transmission owners and regulators are participatory stakeholders. In order to plan necessary upgrades to the electric grid, practitioners first test how anticipated changes would strain the existing electric grid. Only then will operators consider what novel control schemes, metering tools, and grid modernization efforts common in the research literature could be applied to address these concerns.

This research was motivated to answer the question: Will the projected growth in Wisconsin plug-in electric vehicles (PEV) increase electric system congestion and require the construction of new transmission lines? To answer this question, we use the grid modeling tools available to operators, assume no changes to the existing Wisconsin electric grid, and use realistic electric vehicle growth assumptions. This research required access to proprietary electric grid data, not commonly available to researchers for security purposes. This work is written to address the concerns of transmission system practitioners, upon which the research community can build.

Current electric vehicle research considers technology adoption projections and the infrastructure needed to support PEV growth [1-3], and a smaller subset of studies have considered the specific impacts on distribution and transmission electric grids [4-6]. There are few studies specific to the Midwestern United States [3], but no existing work for Wisconsin regarding PEVs and transmission congestion. This study analyzes how electrification of light-duty vehicles will transform the use of the electric power grid along bulk transmission lines. This work considers how projected electric vehicle growth in the State of Wisconsin would impact transmission congestion, using wholesale electricity pricing as a proxy for system impacts. The objective of this research is to create a better understanding of how an emerging PEV market can impact transmission system need.

2) For materials and methods, there are lot of assumptions made in the study, which has also been stated in the conclusions. But the theoretical basis of some of the assumptions are missing. 

  • Lines 114-115: What was inconclusive, what was the basis of discarding the entries, how many entries were discarded. This has to be explained in detail.

In section 2.1, we revised the statement regarding the inconclusive data to include the total number of inconclusive entries and the reasoning for discarding the entries. Page 5, lines 174-192 now read:

Vehicle owners identify their vehicle make and model on registration materials, and often we found that these identifiers did not match manufacturer specified models. As such, we discarded 325 (13% of original dataset) inconclusive vehicles from the data set since we could not confirm if these vehicles were indeed plug-in EVs. Due to these assumptions, the Wisconsin vehicle registrations are conservative compared to other sources [13]. Vehicles are binned within each of the five utility service territories based on registration zip code. While the majority of all PEV charging occurs overnight in owners’ homes [21], this assumption may not accurately represent the service territory in which every vehicle is frequently charged. Vehicle registrations with the primary owner listed under non-Wisconsin zip codes were evenly distributed into all WI service territories.

  • Line 133-138- After looking at the NREL report (reference 14), i feel that the manusricpt needs to justify using the load profiles mentioned in that study. The study has taken into consideration the commercial vehicle charging as well. In the manusricpt there was no mention of the number of commercial charging stations in the targeted region. The results which point to the charging of cars in the afternoon could well be related to the commercial charging station available in a different location as compared to the residential address. Which in my opinion creates doubts about the reliability of the results.

Thank you for pointing this out. We have added Figure 1b which shows the volume and location of public charging stations in Wisconsin. We have additionally added the following to Section 2.1 (page 6, lines 198-201) to discuss the charging profile assumptions in the report and how this relates to Wisconsin. The added language reads:

The EVI-Pro Analysis used by NREL to simulate charging profiles attempts to optimize vehicle charging with operating costs and assumes drivers have access to multiple charging stations throughout the day. The location of nearly 600 public charging stations in the State of Wisconsin [13] are shown in Figure 1(b). The charging stations closely match the location of registered PEVs.

  • The growth scenarios considered, also need to be re-thought in my opinion to give more relatable results.

These growth scenarios were chosen in consultation with the Public Service Commission of Wisconsin to answer questions they posed to the researchers. Like the Department of Transportation data, we do not have the authority to choose different growth scenarios. We hope the editor and reviewer will understand this constraint. We believe this work has value in exposing what questions are of importance to regulators and transmission planners, which we tried to better address in the introductory paragraphs.

  • Line 160-168 - The baseline scenario discussed dosen't mention the number of PEV's it just takes into consideration the load. It refers to explaination in the next section, but i am not sure which one. Exact reference should be put. Secondly for the reference scenario, 0.19% growth rate is not suitable. The rate of adoption should be compounded the inevitable growth of the PEVs, suggested in so many studies in the literature should be considered. This has been done for high adoption when the predicted national adoption rate (it should considered as 5.1% not 5%, mention only the EIA figure or AEO2019 figure) has been used. Also, the figure of 7.48 million in line 167, need to have a basis which should be explained.

Section 2.2 has been revised to discuss the PEV growth scenarios and the vehicle estimate calculations. We hope that this is more understandable. Page 7, lines 232-243 now read:

The baseline scenario uses internal load growth assumptions of the ABB PROMOD software, described in the Section 2.3. These load growth assumptions use proprietary industry data accepted by transmission system planners. We did not add any new electric vehicle charging loads to the PROMOD load assumptions for the baseline scenario.

The reference growth scenario used here was determined using the EIA AEO2019 reference case assuming Wisconsin’s adoption of PEVs continues to maintain a 0.19% share of PEVs in the U.S. in 2030. Wisconsin is assumed to share the same percentage of PEVs in 2030 and grow at the same rate as the U.S. on average. Given these assumptions, the reference growth scenario used in this study results in 26,600 electric vehicles registered in Wisconsin by 2030. Wisconsin currently has a 2.75% share of the U.S. light duty vehicle stock. Assuming Wisconsin maintains the same percentage of U.S. vehicles, total vehicles for 2030 in Wisconsin is estimated to be 7,481,124. Assuming a light-duty vehicle stock of 7.48 million, PEVs would only make up 0.36% of all Wisconsin LDVs in 2030.

  • In the simulation, the authors need to explain what developments (such as steps by the energy suppliers to stabilize the grid, add capacity etc)  does the simulation take into consideration or it takes no developments into account. If it takes the developments in to account, then it would account for the low rise in congestion even in case of high adoption rate. If it dosen't take into account the development scenarios, then the simulation results cannot be treated correct, as assuming that there will be no development in the next 10 years is not suitable. 

Thank you for this comment. The transmission upgrades considered in this study are those announced by all utility and transmission owners in the MISO region. We have rewritten Section 2.3 to better describe the transmission planning assumptions. This section (page 8, lines 261-282) now reads:

To simulate the regional transmission grid, we used ABB’s PROMOD IV (version 11.1) tool. PROMOD is an industry-standard software used by system planners today. MISO uses this tool for production cost simulation modeling in their annual transmission planning process [23]. PROMOD makes general assumptions about utility load and growth for projected years using both publicly and industry-proprietary available generation, load and transmission system data. Notable assumptions made by the PROMOD model include announced generation additions and retirements, proposed transmission system additions, and simplified load profiles that differ only by weekdays, weekends, seasons, and holidays. The baseline Eastern Interconnect transmission system model was used, considering publicly known changes to which Wisconsin generators and transmission lines would likely be in operation by the year 2030.

A baseline load for 2030 is created by the simulator for each of the five Wisconsin service territories using these assumptions for each utility in 2030. We did not add any PEV charging load to the baseline scenario for 2030. To create the reference and high growth load scenarios for each Wisconsin utility, the baseline load as modeled by PROMOD is increased considering the PEVs daily load charging curves and growth scenarios for each utility territory separately.

The built-in PROMOD load assumptions are considered a good projection for load growth. To confirm that the PROMOD load assumptions match practice, we compared historic PROMOD load assumptions with known Wisconsin load. The ABB PROMOD baseline load assumptions for Wisconsin in 2014 closely matched those calculated in the Strategic Energy Assessment (SEA) by the Public Service Commission of Wisconsin in the same year [24], with the exception that the Northern States Power utility territory in PROMOD does not disaggregate that load in Wisconsin from that in Minnesota. We kept the full NSP territory load – both Wisconsin and Minnesota – in this study.

  • The authors also need to take into account the fact that some of the PEV users might be prosumers, who would use their own generated electricity for charging the PEV's. This could be one of the reasons that even in high adoption rate, the impact on grid or prices wouldn't be that much. 

Anticipated changes in distributed generation in 2030 are included in the utility load projections of the baseline scenario. We assumed all additional PEV in the reference and high adoption scenarios will utilize the electrical grid rather than using personal electricity. This assumption is briefly noted in the aforementioned changes to Section 2.3 and in the newly revised Section 2.2. Section 2.2, page 7 lines 226-231 reads:

There are three PEV growth scenarios used in this study: baseline, reference, and high adoption. All projected vehicles are expected to utilize the electric grid for 100% of their charging needs. In many cases, PEV owners may also choose to install distributed generation resources, such as rooftop solar, which would offset their use of the electric grid for charging. Our assumption puts more strain on the grid than assuming some portion of future PEV owners will use distributed generation for vehicle charging.

  • The policy discussion and conclusions need to address all these issues as well.

Additional assumptions were added to the conclusion section. Page 15, lines 370-378 now read:

This study considers how unmanaged electric vehicle charging growth would impact the existing Wisconsin transmission grid. There will be unforeseen electric grid system upgrades that may lessen the impact of electric vehicle load, including owners’ decisions to install residential distributed generation resources. In a high growth scenario, it is likely that distribution utilities would make system upgrades and incentivize customers to charge at desired times of day to specifically accommodate the increase in charging load. For these reasons, the wholesale pricing increases seen at 5% PEVs in this study may be an overestimate. It is possible that increased congestion pricing due to electric vehicle growth may not been seen until more than 5% of Wisconsinites’ vehicles are charging from the grid

Overall the theme of the paper is interesting but i feel that it needs major improvements, for it to be considered for publication. 

Thank you for taking the time to review this work so thoroughly. We hope that we have addressed all your concerns.

Reviewer 2

Open Review

English language and style

( ) Extensive editing of English language and style required 
(x) Moderate English changes required 
( ) English language and style are fine/minor spell check required 
( ) I don't feel qualified to judge about the English language and style 

Yes

Can be improved

Must be improved

Not applicable

Does the introduction provide sufficient background and include all relevant references?

( )

(x)

( )

( )

Is the research design appropriate?

( )

(x)

( )

( )

Are the methods adequately described?

( )

(x)

( )

( )

Are the results clearly presented?

( )

(x)

( )

( )

Are the conclusions supported by the results?

( )

(x)

( )

( )

Comments and Suggestions for Authors

In general the paper discusses the impact of EV on a specific market. Although there is nothing new in the paper but the information are worthwhile, however, i wish the authors to discuss the results in more detail as the significance of such papers should be in results and discussion 

Thank you for taking the time to review our work. We hope that the edits as outlined above have addressed your concerns.

Round 2

Reviewer 1 Report

The manuscript looks much better now and the authors have made significant changes addressing the comments from the previous version.

In the responses, the authors have acknowledged that there are some limitations which can not be addressed. Also, the authors have pointed that the questions they are addressing, is of importance for regulators and transmission planners.

Hence i would suggest two things.

1) In conclusions or in Policy Discussion, add a few lines about the the importance of results for regulators and transmission planners.

2) Clearly add the limitations of the study at the end, in a section, after conclusion. This is very important in my humble opinion.

Author Response

The additions are included in the Conclusions section under a new Limitations section:   Conclusions   This work is the result of a collaboration between state government and the university, which is unfortunately rare within the academic literature. The study was motivated by concerns held by the Public Service Commission of Wisconsin, the state electric grid regulatory body. The regulator is most concerned with how modest load growth scenarios would impact state electricity prices, without assuming the implementation of new policies. This work exposes the questions of importance to regulators and transmission planners in anticipating changes in wholesale congestion pricing.

Limitations

The Wisconsin vehicle registration data was made available by the Wisconsin Department of Transportation on a one-time basis. The authors do not have access to more recent vehicle registration data. We recognize the gap in time between collecting data, analysis and publishing results as a trait of academia, and regret that the registration data used is a year older than the publication date. The PEV growth scenarios used in this study were chosen in consultation with the PSCW to answer questions they posed to the researchers. Similarly, researchers were asked to use the modeling tools available to Commission engineering staff and not academic-built modeling tools. The use of PSCW tools enabled reputable results that Commission staff can easily interpret and trust. In a purely academic study, more aggressive load growth scenarios, modeling assumptions that include intelligent charging and coupled distributed generation – PEV charging control schema, and publicly reviewed production cost modeling tools may have been used.